# Life Cycle Carbon Emission Assessment of Building Refurbishment: A Case Study of Zero-Carbon Pavilion in Shanghai Yangpu Riverside

**Boyang Li [1], Yiqun Pan [1,*] , Linxue Li [2,*] and Mingshu Kong [3]**

1   School of Mechanical Engineering, Tongji University, Shanghai 201804, China
2   College of Architecture and Urban Planning, Tongji University, Shanghai 200092, China
3   Tongji Architectural Design (Group) Co., Ltd., Shanghai 200092, China
*   Correspondence: yiqunpan@tongji.edu.cn (Y.P.); land_well@hotmail.com (L.L.)

**Abstract:** Life cycle building carbon assessment can promote the development of carbon emission reduction. The main difficulties in the process of assessment are the boundary selection and inventory collection, especially when carbon emission assessment is needed in the early stage of design and construction, or when the calculation relates to disposal and refurbishment. It is significant to make full and rational use of design documents, standard documents, and related software. This paper focuses on the life cycle carbon emission assessment of building refurbishment. It explores the carbon emission assessment methodologies and procedures in every phase of the building life cycle, taking a zero-carbon pavilion refurbishment project as a case study. This case study is located in the Shanghai Yangpu Riverside Park, refurbished from an existing hydrologic monitoring building. The carbon emission reduction potential of renovation and the solar photovoltaic system applied in the building are analyzed. The data was collected referring to architectural design documents and related standards. The energy consumption during the operational phase is simulated using DesignBuilder. The life-cycle carbon emission per floor area of the existing building renovation scenario is 2.39 t, and the new building scenario is 2.69 t, which are both at a low level among other cases. The refurbished existing building saves nearly one-third of the carbon dioxide emissions during the construction phase compared to new construction. The application of a photovoltaic system also saves one-third of energy consumption and carbon emissions during the operational phase.

**Keywords:** life-cycle carbon emission; refurbishment of existing buildings; energy saving





## 1. Introduction

The rapid development of the construction industry has led to high energy consumption and greenhouse gas emissions. According to statistics from the International Energy Agency, the construction industry accounts for 37% of global carbon emissions, and has become one of the key carbon reduction areas [1].

The relevant research is mainly concentrated in developed countries and building carbon emissions in developing countries are gradually gaining more attention. According to Yue Teng's statistical research, China has the most published literature on building carbon emissions [2]. As a rapidly growing developing country, China's energy conservation and emission reduction path will provide ideas for many developing countries in the world [1]. In order to promote building environmental impact assessment more broadly, it is important to establish a building carbon emission database and related standards classified by regions and types of building [3,4]. Referring to relevant research results, the general level of building whole life cycle carbon emissions can be estimated. Bo Peng compiled data for 104 calculation cases; the whole life cycle energy consumption of residential buildings was mainly in the range of 40–400 kWh/(m² a) [5], and the public buildings were in the range of 120–550 kWh/((m²a). Zhang Xiaocun's study collected and compiled 348 cases

of building carbon emission calculations [6], and most of them had carbon emissions in the range of 0.25–0.6 tCO$_2$e/m$^2$ for the production phase of building materials. In this paper, 18 calculation cases in China were selected for statistics. The whole life cycle carbon emission is in the range of 2–5 tCO$_2$e/m$^2$. The materialization phase is generally in the range of 0.3–0.8 tCO$_2$e/m$^2$, accounting for 10–30% of the whole life cycle carbon emission. The operation phase is generally in the range of 2–4.5 tCO$_2$e/m$^2$, accounting for 75–90% of the whole life cycle carbon emission [6–23].

The calculation of building carbon emissions is an important basis for the development of zero carbon buildings. The most common method of carbon emission assessment for single buildings is the life cycle assessment (LCA) [5]. The whole life cycle of a building generally refers to the whole process from the production of building materials to building demolition and disposal. The sum total of greenhouse gas emissions from energy consumption during the materials production, transportation, construction, operation, and disposal phases are known as life cycle carbon emissions. Key steps of life cycle assessment methodology include the identification of goals and scope, inventory analysis, impact assessment, and interpretation of results [24].

The calculation methods for a single buildings' carbon emission assessment have been developed gradually. The existing research mainly focuses on the calculation framework and boundary, calculation tools and the source of uncertainty in the calculation process. However, whole life cycle carbon emission is often needed to be known at an early stage of the building design to obtain more energy-efficient design solutions. Roberts pointed out in a literature review that few studies are focusing on carbon emissions calculations in the absence of actual documented data and other information, and that there are great challenges in the selection of data collection processes [25]. The difficulties are as follows. First, life cycle assessments are mostly based on the assumed stable state. However, the users' behavior, energy consumption, and external environment are in the process of dynamic change and reliable dynamic assessment methods are still lacking. Second, the data resource varies from different life cycle stages, the data sources and computational boundary selection rules are inconsistent among different cases. The third is about the application of computational tools, which mainly include LCA plug-ins in BIM software and specialized life cycle computing software to import building information and export the calculation results. Reginal templates and default parameters in software tools can be easily applied as supplements in the early stages of architectural design, while also simplifying the users' input. However, Thais Sartori [26] found that many widely used LCA algorithms are in a "black box" state for intellectual property protection and user-friendliness of developers, making the process less transparent and more difficult for users to interpret, hindering the search for the optimization of building performance [27]. In summary, it is worthwhile to investigate how to choose the right data sources and calculation model, while using software and relevant standard documents as supports and supplements.

Nowadays, as the proportion of existing buildings rises rapidly, the energy consumption of buildings is increasing. Consequently, it is significant to discuss the carbon emissions generated before and after refurbishment. Many studies began to pay attention to the renovation of existing buildings and the environmental impact during renovation. For example, Nihat Atmaca compared the whole life cycle energy consumption of new and renovated heritage buildings based on a case study [28]. Usha Iyer-Raniga analyzed the energy-saving potential of different heritage buildings using LCA and simulation software, and proposed energy-saving and emission reduction measures that could be taken during the construction, operation, and maintenance phases [29]. However, there are few studies systematically discussing carbon emission calculation methods for building refurbishment. Distinguished from ordinary new construction, it is more difficult for refurbished buildings to define the boundary and assess the uncertainty during calculation. It is also worth exploring how to balance the environmental and social impacts of building renovation [23].

This paper mainly explores the building life cycle carbon emission assessment methodologies of refurbished buildings in the early stage of architectural design. The energy

consumption of each phase is analyzed, and the generated carbon emissions are summed up to obtain the whole life cycle carbon emissions. A zero-carbon pavilion refurbishment project is taken as a case study, in which various energy saving and carbon reduction measures are applied.

## 2. Materials and Methods

### 2.1. The Calculation Model of Building Whole Life Cycle Carbon Emission

The whole life cycle of a building can be divided into the building materials production phase, materials transportation phase, building construction phase, building operational phase, and demolition phase. The whole life cycle carbon emission is the sum total of carbon emissions of the five phases, which can be expressed by the following equation [30].

$$C_{\text{(Whole Life cycle)}} = C_{\text{Production}} + C_{\text{Transportation}} + C_{\text{Construction}} + C_{\text{Operation}} + C_{\text{Demolition}}$$

Meanwhile, the data source varies among different life cycle phases of building. The most ideal situation is that all the data used in the calculation are from actual engineering record documents. However, the building may be in the phases of design, construction, or early operation, which means varying degrees of data deficiency. Different data sources and paths can be chosen according to the actual situation. Figure 1 shows the calculation frame model, and optional data sources and corresponding data processing solutions [31].

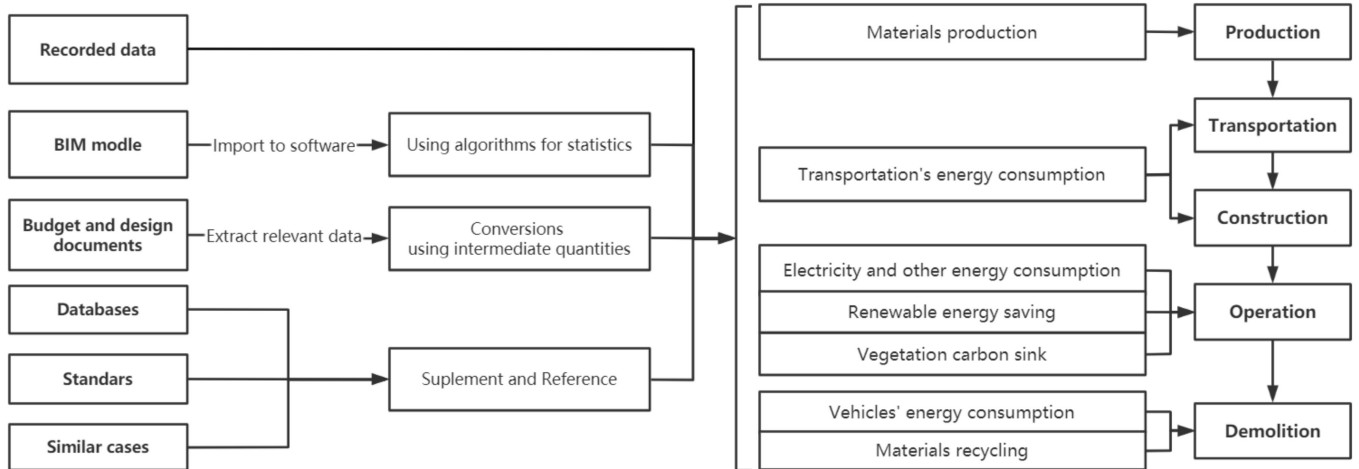

**Figure 1.** Frame of building whole life cycle carbon emission calculation model.

The carbon emission factor method is used to calculate the carbon emission of each phase. There are three main forms in practical applications.

1.  For the carbon emissions generated directly from energy consumption:

$$C_{\text{Energy}} = E_i \, F_{Ei}$$

$C_{\text{Energy}}$: The carbon emissions generated by energy consumption
$E_i$: Energy consumption
$F_i$: Carbon emission factor (carbon emission per unit of energy consumption)

2.  For the carbon emissions generated from materials production:

$$C_{\text{Materials}} = M_i \, F_{Mi}$$

$C_{\text{Materials}}$: The carbon emissions generated by materials production
$M_i$: Amount of material weight, length, surface area, volume, etc.)
$F_{Mi}$: Carbon emission factor (carbon emission per unit amount of material)

3. For the carbon emissions generated from engineering processes:

$$C_{\text{Engineering process}} = Q_i\, F_{Pi}$$

$C_{\text{Engineering process}}$: The carbon emissions generated during the engineering process
$Q_i$: Quantity of constructure work (length, surface area, volume, time, number of machines, etc.)
$F_{Pi}$: Carbon emission factor (carbon emission per unit of engineering work)

### 2.2. Introduction of the Case Study

The building of the case study in this paper is refurbished from an existing hydrologic monitoring building, which will be used as a museum for science exhibitions. It is located in the Shanghai Yangpu Riverside Park and is close to the Huangpu River, surrounded by several industrial historical buildings. Its geographical location and surrounding profile are shown in Figure 2.

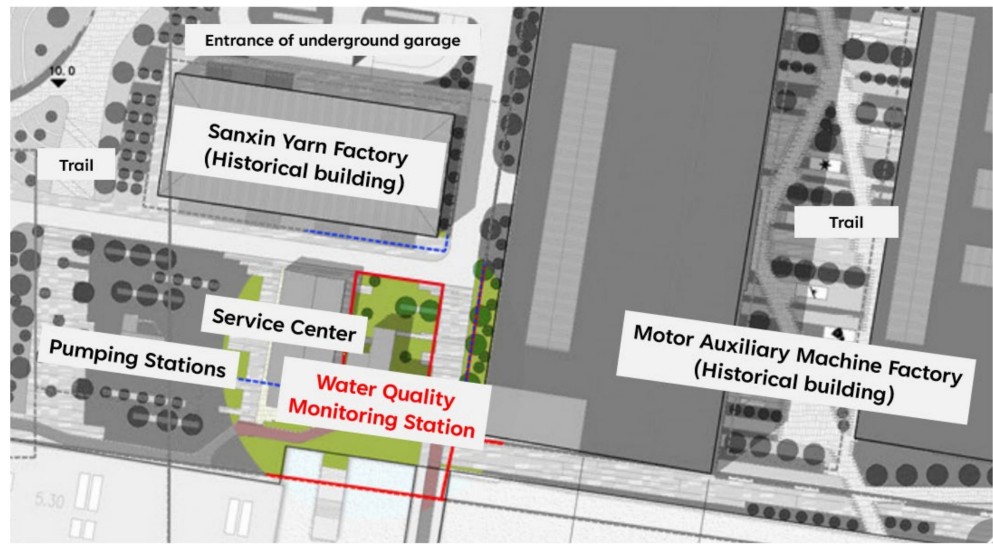

**Figure 2.** The geographical location of the water quality monitoring station.

The building consists of three main components: an above-ground monitoring station, an underground exhibition hall, and the surrounding landscape. Table 1 and Figures 3 and 4 show detailed information about the building.

**Table 1.** Detailed information about the studied case.

| Project Name | Yangpu Bridge Public Space and Comprehensive Environment Project—Water Quality Monitoring Station Refurbishment |
|---|---|
| Project address | Yangpu district of Shanghai |
| Climate zones | Hot-summer and cold-winter zone |
| Building classification | Public buildings (Class A) |
| Structure type | Masonry structure |
| Floor area | 467 m$^2$ |
| Existing floor area above ground | 65 m$^2$ |
| Expansion floor area above ground | 43 m$^2$ |
| Underground floor area | 360 m$^2$ |
| Building height | 4.997 m |
| Building depth | 4.7 m |

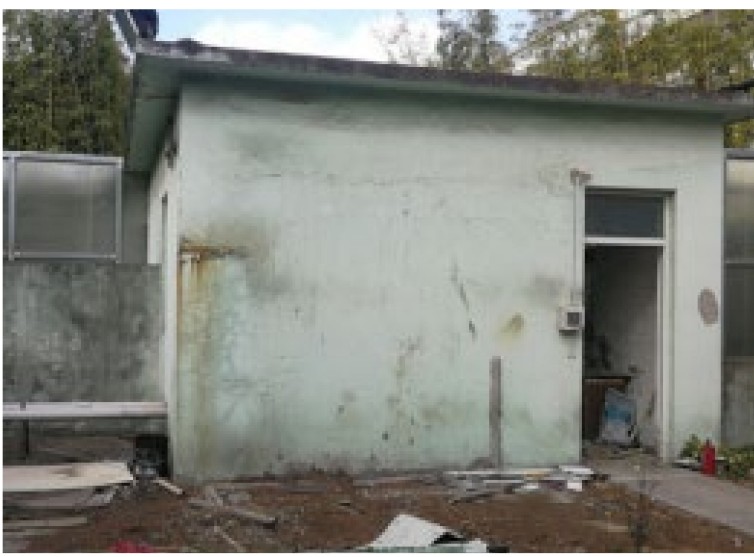

**Figure 3.** The old building of the water quality monitoring station.

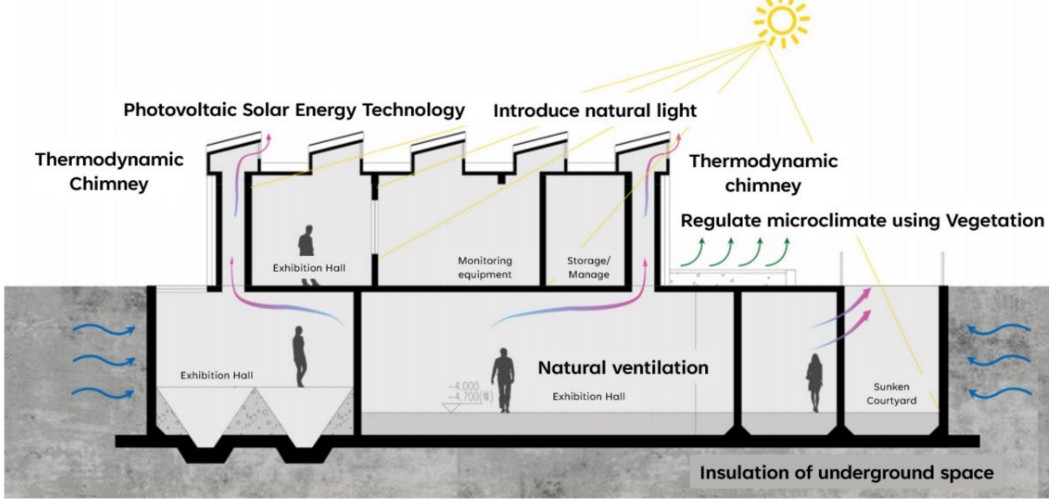

**Figure 4.** Building diagram of Zero-Carbon Pavilion in Shanghai Yangpu Riverside.

Shanghai belongs to the hot summer and cold winter region. To ensure the comfort of users, cooling in summer and heating in winter may cause a great amount of energy consumption. The project adopts various energy-saving and carbon emission reduction measures, including introducing natural light into an underground space, using thermal insulation in underground space, and creating thermodynamic chimneys by opening holes in the roof. Several active technologies have also been applied, including the use of air-cooled heat pumps for cooling and heating sources and radiant floors in exhibition areas, the installation of solar photovoltaic systems for exterior landscaping and roofs, and the application of hydrogen energy storage technologies.

To analyze the energy-saving potential of building refurbishment compared with new construction, a hypothetical new building will be studied together. The geographical location, structure, and functions are the same as the refurbished building. The only difference between them is the construction process. The construction process of the new building uses new materials and refers to the general construction process adopted by existing engineering projects. The information about the old building is mostly from actual design documents.

### 2.3. Calculation Scheme and Assumption

The LCA and the carbon emission factors method are used to calculate the carbon emissions of the case building. As it is in the constructure stage, the data of materials and the construction process were collected from architectural design and budget documents. Related standards and databases are referred to as supplements. The energy consumption during the operational phase is simulated using DesignBuilder, in which a virtual model is established.

In addition, specific assumptions and instructions to be made in the calculation process are as follows.

1.  This paper uses $kgCO_2e$ and $tCO_2e$ as the units to calculate the whole life cycle carbon emissions.
2.  The area to be used in calculating of carbon emissions per unit area is 717 $m^2$, which is the sum of building area and landscape area.
3.  The carbon emission factors of construction materials and energy consumption come from the "Building Carbon Emission Calculation Standard", which includes the equivalent environmental impact caused by the other greenhouse gases.
4.  The electricity consumption carbon emission factor used in this paper is 4.2 $tCO_2/10^4$ kWh, according to the notice of Shanghai Municipal Bureau of Ecology and Environment on the "Adjustment of the Values of Emission Factors Related to the City's Greenhouse Gas Emission Accounting Guidelines", issued in February 2022 [32].
5.  The new building has the same operation carbon emission as the refurbished building.
6.  It is assumed that the service life of the building is 50 years, and the constructure materials are the same as the whole building [28]. To simplify the calculation, the total energy consumption and carbon emission of 50 years are based on the one-year situation result in the software.

## 3. Results

### 3.1. Carbon Emission Calculation of the Case Study

#### 3.1.1. Building Materials Production

The data sources of the construction materials used for refurbishment are the "Construction tender quotations for overground landscape and underground space refurbishment of Zero-Carbon Pavilion in Shanghai Yangpu Riverside ". The list of building materials is based on the budgets of each sub-project. The summary of carbon emissions in the production phase of refurbished buildings is shown in Table 2.

**Table 2.** Materials list of the refurbished building.

| Building Materials | | Carbon Emission Intensity | Unit | Quantity | Unit | Carbon Emissions ($kgCO_2e$) |
|---|---|---|---|---|---|---|
| Steels | Steel structure components | 2050 | $kgCO_2e/t$ | 19.31 | t | 39,575.65 |
| | Rebar | 2340 | $kgCO_2e/t$ | 28.96 | t | 67,770.63 |
| Concrete | Ready-mixed concrete (non-pumping type) C35 | 335 | $kgCO_2e/m^3$ | 227.79 | $m^3$ | 76,309.63 |
| | Ready-mixed concrete (non-pumping type) C25 | 248 | $kgCO_2e/m^3$ | 24.64 | $m^3$ | 6111.33 |
| Concrete blocks | Concrete brick | 336 | $kgCO_2e/m^3$ | 6.73 | $m^3$ | 2260.83 |

**Table 2.** *Cont.*

| Building Materials | | Carbon Emission Intensity | Unit | Quantity | Unit | Carbon Emissions (kgCO$_2$e) |
|---|---|---|---|---|---|---|
| Mortar | Dry mixed plastering mortar DP M15.0 | 298.73 | kgCO$_2$e/m$^3$ | 14.17 | m$^3$ | 4234.47 |
| | Dry mixed plastering mortar DP M20.0 | 466.35 | kgCO$_2$e/m$^3$ | 5.86 | m$^3$ | 2731.81 |
| | Dry mixed plastering mortar DS M15.0 | 340.44 | kgCO$_2$e/m$^3$ | 21.23 | m$^3$ | 7229.11 |
| | Dry mixed plastering mortar DS M20.0 | 403.46 | kgCO$_2$e/m$^3$ | 15.31 | m$^3$ | 6177.45 |
| Waterproof, heat insulation materials | Asphalt waterproof coil | 4.01 | kgCO$_2$e/m$^2$ | 4005.42 | m$^2$ | 16,061.75 |
| | Squeeze polystyrene board | 4620 | kgCO$_2$e/t | 9.56 | t | 44,162.12 |
| Glass | Flat glass | 1130 | kgCO$_2$e/t | 3.46 | t | 3905.70 |
| Aluminum alloy | Broken bridge aluminum alloy window | 194 | kgCO$_2$e/m$^2$ | 7.28 | m$^2$ | 1412.32 |
| Ballast | Gravel | 2.18 | kgCO$_2$e/t | 120.08 | t | 261.77 |
| Solar photovoltaic panels | Solar photovoltaic panels | 1809.47 | kgCO$_2$e/kW | 8.29 | kW | 15,003.62 |
| Total carbon emission | | 293,208.17 kgCO$_2$e | | | | |
| Carbon emissions per unit area | | 408.94 kgCO$_2$e/m$^2$ | | | | |

Since the type and quantity of building materials of the assumed new building are unknown, the statistic result based on the architectural geometry model constructed in the DesignBuilder software is exported for calculation. The geometry model is shown in Figure 5.

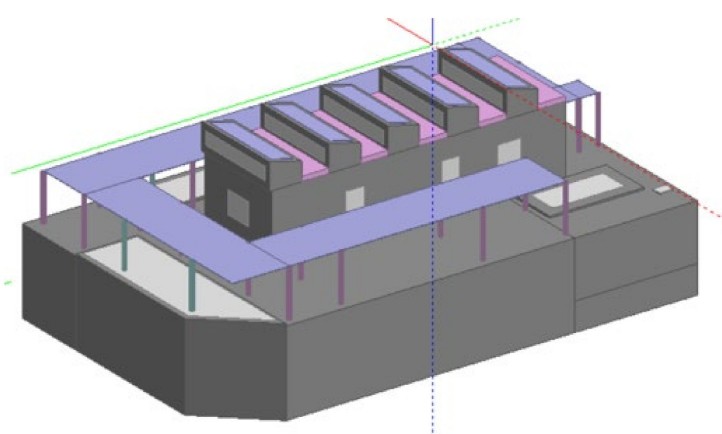

**Figure 5.** Architectural geometric model.

The carbon emission intensity in the production phase of solar photovoltaic systems refers to existing case studies [33–37]. The average value is 8971.79 kgCO$_2$e/kW, which is suitable for both refurbishment construction and new construction, and the power generation of the photovoltaic system is derived from software statistics.

The selection of carbon emission factors for building materials refers to the Building Carbon Emission Calculation Standard.

### 3.1.2. Building Materials Transportation

The weight of building materials in the transportation phases is from the statistical data of the material production phases. The transportations that are selected refer to the "Building Carbon Emission Calculation Standard". Considering transport efficiency, the carrying capacity of the transportations is matched with the quantity of each of the building materials. The default transport distance is 40 km for concrete and 500 km for other building materials [30]. The carbon emission factors of the transport vehicles are referred to as the "Building Carbon Emission Calculation Standard".

### 3.1.3. Refurbishment and Construction

The construction process is summarized and sorted referring to the budget document "Construction tender quotations for overground landscape and underground space refurbishment of Zero-Carbon Pavilion in Shanghai Yangpu Riverside ". The types of equipment and the number of classes used in each engineer process shall refer to the "Shanghai Construction and Decoration Project Budget Quota" [38] and "Shanghai Housing Construction Engineering Maintenance and Repair Budget Quota, Book I" [39], issued by the Shanghai Municipal Commission of Housing and Urban-rural Development. The carbon emission factors of the equipment refer to the "Building Carbon Emission Calculation Standard". The calculation details are shown in Table 3.

**Table 3.** Construction carbon emission of refurbished buildings.

| Vehicles | Energy Consumption | Carbon Emission Factor | Carbon Emission (kgCO$_2$e) |
|---|---|---|---|
| Mortar, concrete mixer | 740.4 kWh (Electricity) | 0.42 kgCO$_2$e/kWh | 5237.09 |
| Electric rammer | 33.2 kWh (Electricity) | 0.42 kgCO$_2$e/kWh | 6.04 |
| Flat water polisher | 56 kWh (Electricity) | 0.42 kgCO$_2$e/kWh | 436.02 |
| Electric air compressor | 80.6 kWh (Electricity) | 0.42 kgCO$_2$e/kWh | 499.25 |
| Air hammer | 169.4 kWh (Electricity) | 0.42 kgCO$_2$e/kWh | 360.00 |
| Steel bar extrusion link machine | 15.94 kWh (Electricity) | 0.42 kgCO$_2$e/kWh | 18.30 |
| AC arc welder | 193.06 kWh (Electricity) | 0.42 kgCO$_2$e/kWh | 254.65 |
| On-site transportation | 325.992 t, 0 km | 0.06 kgCO$_2$e/(t*km) | 781.87 |
| Total carbon emission | | 7593.23 kgCO$_2$e | |
| Carbon emissions per unit area | | 10.59 kgCO$_2$e/m$^2$ | |

As for new construction, due to the lack of engineering documents, this paper lists the common energy consumption of each process, and the carbon emission of this project is estimated based on the information in budget documents [2,21]. The calculation process is described in Table 4.

**Table 4.** Construction carbon emission of new buildings.

| Engineering Process | Engineering Quantity | Unit | Energy Consumption Per Unit of Engineering Quantity (kWh) | Carbon Emission Factor | Carbon Emission (kgCO$_2$e) |
|---|---|---|---|---|---|
| Premixed concrete | 409.35 | t | 25 | 0.42 kgCO$_2$e/kWh | 4298.23 |
| Excavate and remove the earthwork | 2260.8 | m$^3$ | 32 | 0.42 kgCO$_2$e/kWh | 30,385.15 |
| Flat earthwork | 292.5 | t | 3 | 0.42 kgCO$_2$e/kWh | 368.55 |
| Crane handling | 717 | m$^2$ | 2 | 0.42 kgCO$_2$e/kWh | 602.28 |
| Lighting | 717 | m$^2$ | 26 | 0.42 kgCO$_2$e/kWh | 7829.64 |
| Total carbon emission | | | 43483.85 kgCO$_2$e | | |
| Carbon emissions per unit area | | | 60.65 kgCO$_2$e/m$^2$ | | |

### 3.1.4. Building Operation Phase

The energy consumption of the building operation phase is obtained from the Design-Builder software, in which the building geometry and HVAC system model are constructed. At the same time, the users' activities, envelope, equipment parameters, and solar photovoltaic system parameters are also set according to the construction instructions. The composition and heat transfer coefficient of each envelope in the software are summarized in Table 5, and the parameters of each partition and air conditioning systems are summarized in Table 6. The carbon sink data of green vegetation comes from the budget document. According to the design drawings and construction specifications, the project is designed to have a service life of 50 years.

**Table 5.** Composition and heat transfer coefficient of the envelope.

| Envelope | Composition | Name of Materials | Thickness (mm) | Heat Transfer Coefficient (W/m²·K) |
|---|---|---|---|---|
| Above-ground exterior wall | Finishing layer<br>Plastering layer<br>Plinth<br>Insulating layer<br>Interior decoration layer | Coating material<br>Anti-crack waterproof mortar<br>Aerated concrete blocks<br>Rock wool board<br>Mixed mortar | 5<br>200<br>60<br>15 | 0.52 |
| Interior wall | Interior decoration layer<br>Plinth<br>Interior decoration layer | Mixed mortar<br>Concrete block<br>Mixed mortar | 5<br>200<br>5 | 2.15 |
| Roofing | Protection layer<br>Leveling layer<br>Slope finding layer<br>Insulating layer<br>Structural layer | Fine aggregate concrete<br>Cement mortar<br>Cement(-sand) mortar<br>Squeeze polystyrene board<br>Steel-concrete structure roof panel | 50<br>20<br>50<br>60<br>120 | 0.47 |

**Table 6.** Parameters of air conditioning system and other equipment.

| | Partitions | Temperature Summer/ Winter (°C) | Fresh Air Quantity | Personnel Density (p/m²) | Air Conditioning System | Lighting Power Consumption (W/m²) | Equipment Power Consumption (W/m²) |
|---|---|---|---|---|---|---|---|
| Over ground | Exhibition area | 24/21 | 30 m³/ (h·p) | 0.35 | Air-cooled heat pump + Radiant floor + Fresh air system | 40 | 18 |
| | Experimental area | 25/22 | 43.2 m³/ (h·p) | 0.097 | Packaged air conditioning unit | 43 | 20 |
| | Administrative area | 25/21 | 36 m³/ (h·p) | 0.0987 | Packaged air conditioning unit | 32 | 20 |
| | Explosion-proof area | | 12 times/h | 0.1238 | Mechanical exhaust | | |
| Under ground | Exhibition area | 24/21 | 30 m³/ (h·p) | 0.35 | Air-cooled heat pump + Radiant floor + Fresh air system | 40 | 18 |
| | Equipment room | | 12 times/h | 0.1238 | Mechanical exhaust | | |
| | Toilet | | 10 times/h | 0.1238 | Mechanical exhaust | 200 lux | |

There are two main energy-saving measures during the operational phase. The air conditioning system of the exhibition area consists of an air-cooled heat pump and a radiant floor. Its air conditioning system model is shown in Figure 6. The solar photovoltaic panels are installed on the roof of the building and the surrounding corridor and then access to the power supply system. Mechanical and equipment parameters are shown in Table 7. The energy consumption of the operations phase based on software simulation is shown in Figure 7.

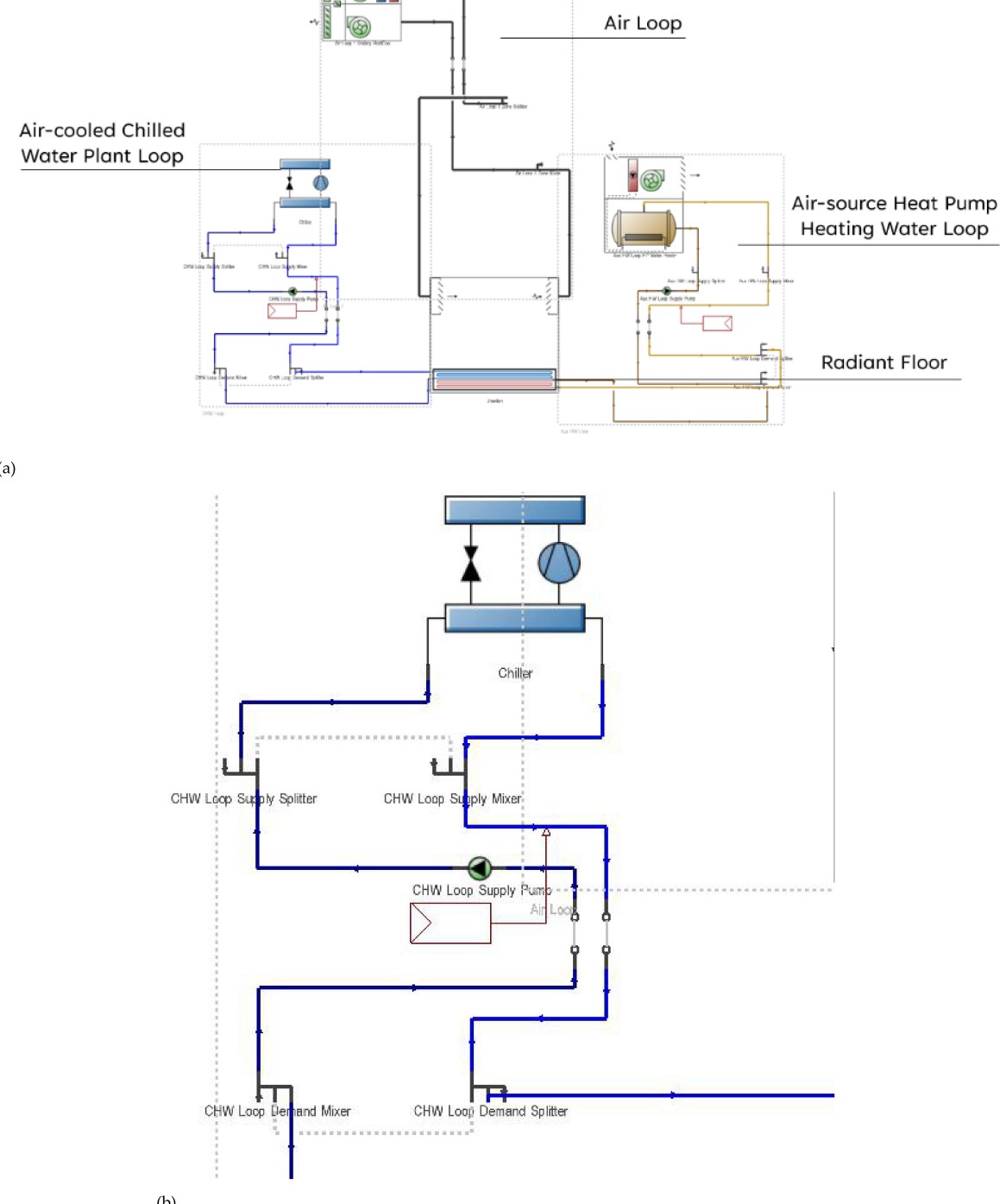

(a)

(b)

**Figure 6.** *Cont.*

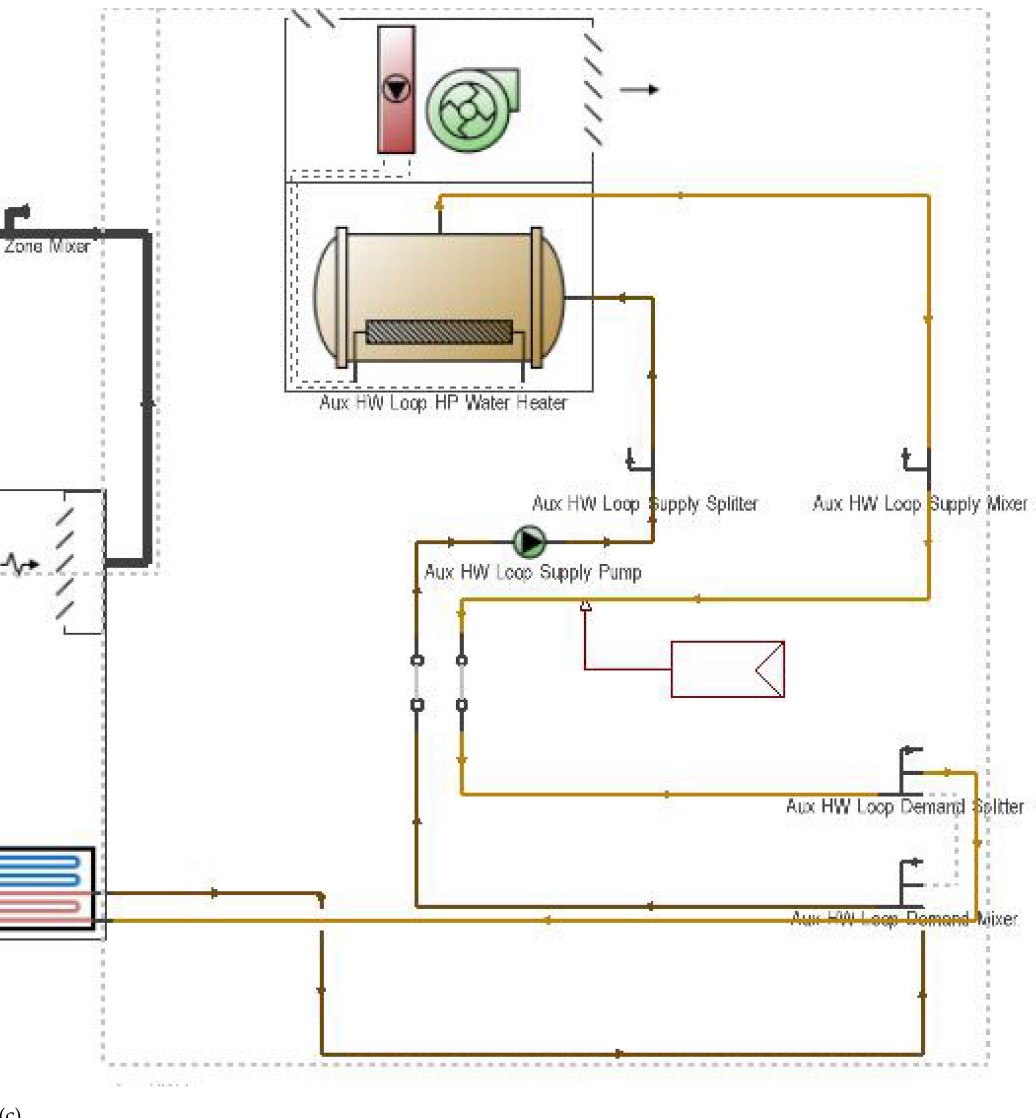

(c)

**Figure 6.** *Cont.*

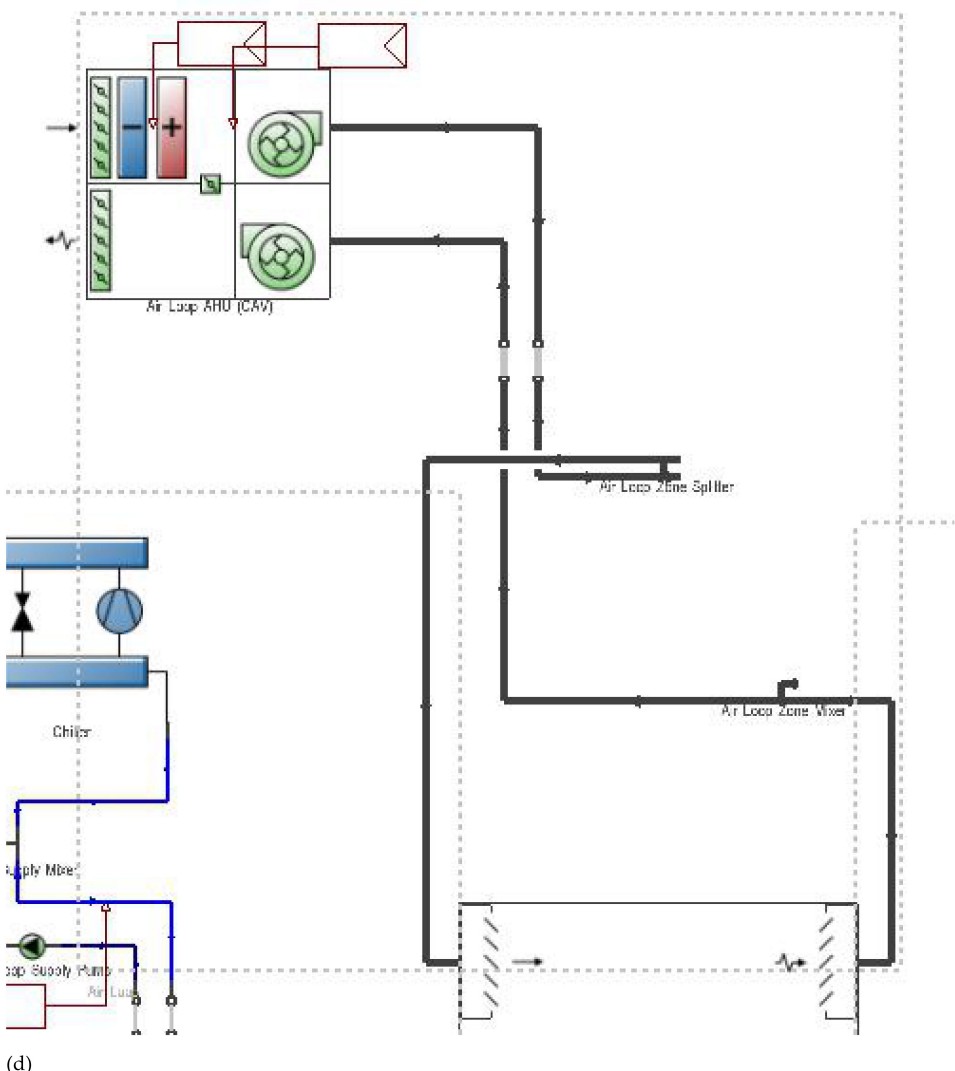

(d)

**Figure 6.** Air-conditioning system model in the exhibition area: (**a**) The whole air-conditioning system. (**b**) Air-cooled Chilled Water Plant Loop. (**c**) Air-source Heat Pump Heating Water Loop. (**d**) Air Loop.

**Table 7.** Mechanical and equipment parameters.

| Equipment | Equipment Parameters | Value |
|---|---|---|
| Split air conditioners | Heating capacity<br>Cooling capacity | 3.8 kW<br>3.5 kW |
| Air-cooled water chiller<br>Air-source heat pump | Cooling capacity<br>Heating capacity | 13.20 kW<br>11.02 kW |
| Fresh air unit | Cooling capacity<br>Heating capacity | 10.45 kW<br>11.42 kW |
| Fans | Air volume<br>External static pressure | 350 CMH<br>50 Pa |
| Solar photovoltaic panel | Efficiency | 0.15 |

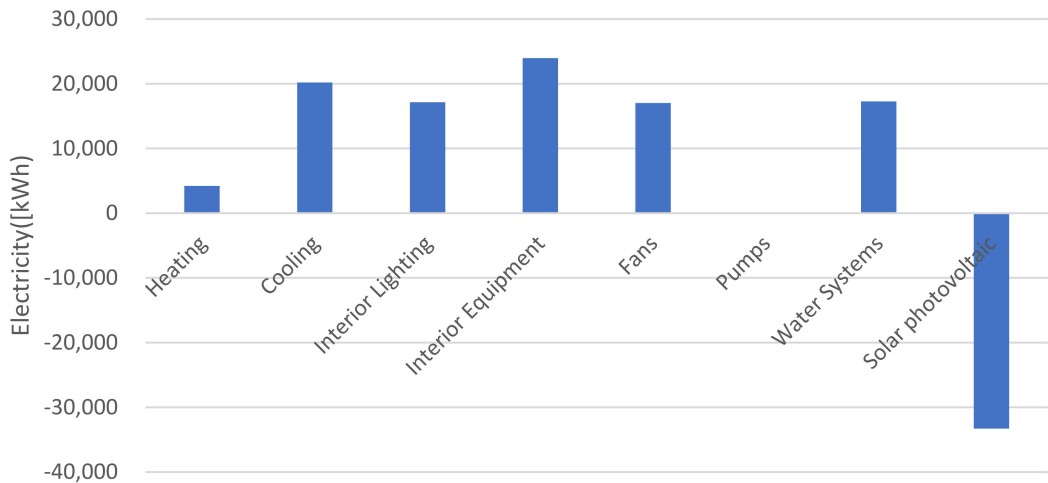

**Figure 7.** Comparison of power consumption and power generation of building operation.

3.1.5. Building Demolition Phase

Due to the lack of actual engineering data during the demolition phase, the general energy consumption was summed up based on existing cases [5,23]. Other data comes from engineering design documents. Calculation details are shown in Table 8.

**Table 8.** Carbon emission calculation during the demolition phase.

| Engineering Process | Engineering Quantity | Unit | Energy Consumption Per Unit of Engineering Quantity (kWh) | Carbon Emission Factor | Carbon Emission (kgCO$_2$e) |
|---|---|---|---|---|---|
| Component removal | 717.00 | m$^2$ | 29.5 MJ | 0.42 | 2467.68 |
| Flat earthwork | 717.00 | m$^2$ | 7.2 MJ | 0.42 | 602.28 |
| Crane handling | 1086.85 | t | 10.8 MJ | 0.42 | 1369.44 |
| Total carbon emission | | | 4439.39 kgCO$_2$e | | |
| Carbon emissions per unit area | | | 6.19 kgCO$_2$e/m$^2$ | | |

*3.2. Calculation Result*

3.2.1. Summary of the Calculation Result

The whole life cycle carbon emissions of the building and the proportion of each phase are shown in Table 9 and Figure 8. It can be seen from the chart that the carbon emissions of the building operation phase occupy the highest proportion in the whole life cycle, followed by the building materials production phase, while the carbon emissions of the building materials transportation, building construction, and building demolition phases account for a small proportion.

**Table 9.** Whole life cycle carbon emissions of refurbished building and new building.

| Phases | Refurbished Building | | New Building | | Total Emission Reduction (kgCO$_2$e) | Emission Reduction Ratio | The Proportion of Emission Reduction |
|---|---|---|---|---|---|---|---|
| | Total (kgCO$_2$e) | Per Unit (kgCO$_2$e/m$^2$) | Total (kgCO$_2$e) | Per Unit (kgCO$_2$e/m$^2$) | | | |
| Materials production | 293,208.17 | 408.94 | 459,716.67 | 641.17 | 122,346.38 | 36.22% | 77.97% |
| Materials transportation | 13,626.07 | 19.00 | 24,782.04 | 34.56 | 11,155.97 | 45.02% | 5.22% |

**Table 9.** *Cont.*

| Phases | Refurbished Building | | New Building | | Total Emission Reduction (kgCO₂e) | Emission Reduction Ratio | The Proportion of Emission Reduction |
|---|---|---|---|---|---|---|---|
| | Total (kgCO₂e) | Per Unit (kgCO₂e/m²) | Total (kgCO₂e) | Per Unit (kgCO₂e/m²) | | | |
| Building operations | 7593.23 | 10.59 | 43,483.85 | 60.65 | 35,890.62 | 82.54% | 16.81% |
| Building operation | 1396,691.69 | 1947.97 | 1,396,691.69 | 1947.97 | 0 | 0.00% | 0.00% |
| Building demolition | 4439.39 | 6.19 | 4439.39 | 6.19 | 0 | 0.00% | 0.00% |
| Total carbon emission | 1,715,558.56 | 2392.69 | 1,929,113.65 | 2690.54 | 213,555.09 | 11.07% | 100.00% |

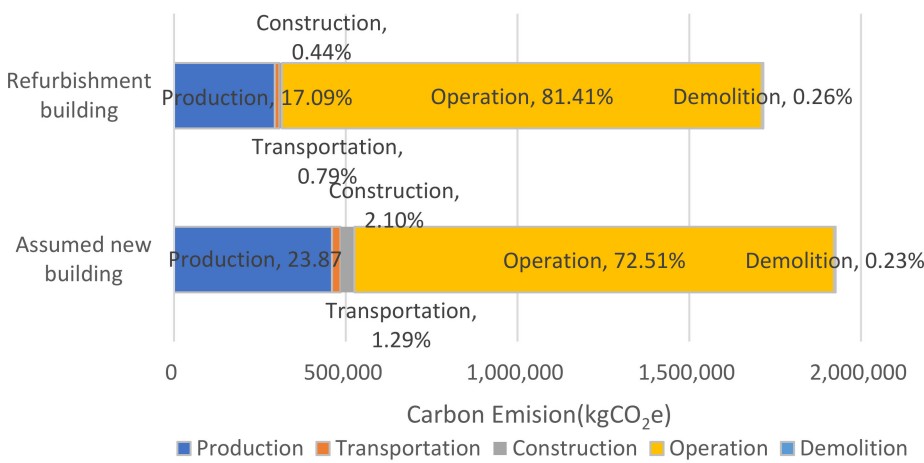

**Figure 8.** Carbon emission of refurbishment building and new building and proportion of each phase in the whole life cycle.

### 3.2.2. Carbon Reduction of Building Refurbishment Compared to New Construction

Compared with the new construction, the whole life cycle carbon emissions of the refurbished building are reduced by 213.56 tCO₂e, accounting for 11.07% of the total carbon emissions of new buildings, and the carbon emission reduction of the production phase of building materials accounted for 77.97% of the total reduction. The carbon emission of major building materials is analyzed separately for comparison, as shown in Figure 9. The carbon emission reduction of major building materials, such as steel, concrete, cement mortar, clay, and insulation materials, reached 118.93 tCO₂e, accounting for 71.42% of the overall carbon emission reduction of building materials. In the process of building refurbishment, the original foundation of the refurbished building offsets the carbon emissions of building materials that may be generated by the construction of new buildings, avoiding the repeated emissions caused by reconstruction after demolition.

### 3.2.3. Analysis of Carbon Emission Reduction Benefits of the Solar Photovoltaic System

The changes in carbon emissions before and after the application of the solar photovoltaic system are shown in Figure 10. As can be seen from the figure, the carbon emissions reduction during the operation phase is much greater than the carbon emissions generated by its production, transportation, and installation phases.

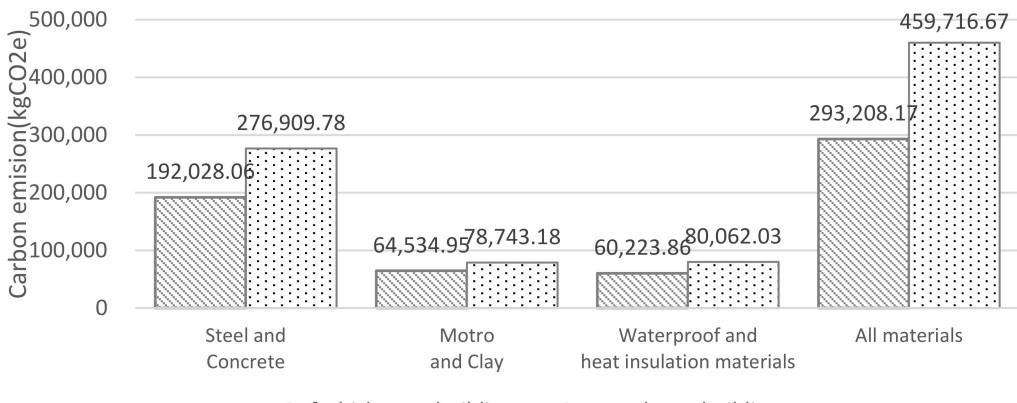

**Figure 9.** Carbon emission from the production of the main building materials.

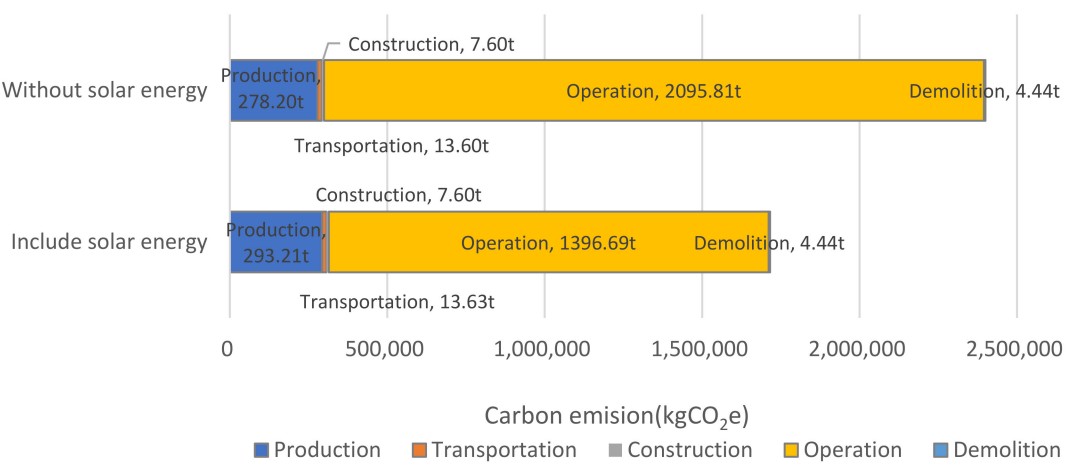

**Figure 10.** Carbon emissions before and after the solar photovoltaic system applied in refurbishment building.

### 3.3. Assessment of the Whole Life Cycle Carbon Emission Level of Buildings

Based on the statistical results of the 18 China cases in the introduction, the carbon emission of the building can be estimated to be 2–8 $tCO_2e/m^2$ in the whole life cycle, 0.3–0.8 $tCO_2e/m^2$ in the construction phase, and 2–4.5 $tCO_2e/m^2$ in the operation phase. The refurbishment building and the assumed new building are both at a lower carbon emission level among the public calculation cases. However, for the latter, the carbon emission of the construction phase is at a high level. The comparison results are shown in Figure 11 and Table 10.

Most of these cases have actual recorded materials. Additionally, the carbon emission factors in these cases are from similar references, such as the "Building Carbon Emission Calculation Standard" issued by the Ministry of Housing and Urban-Rural Construction of the People's Republic of China. However, it is worth noting that, due to the inconsistency of calculation methods and data resources, the statistical result is only used to reflect the carbon emission level of the case study roughly and assess the reliability of the calculation process.

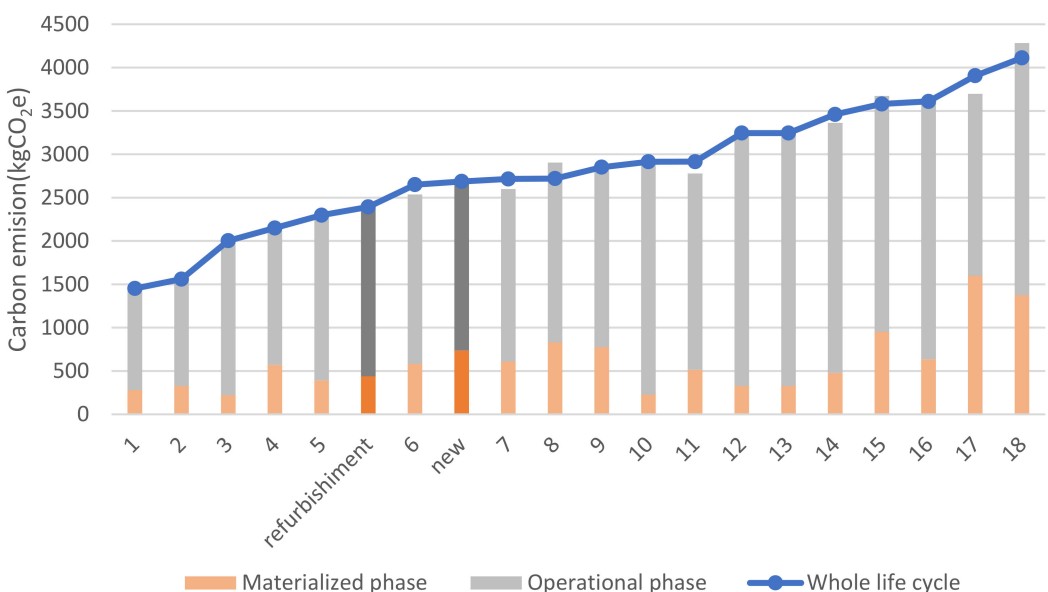

**Figure 11.** Comparison between existing cases and this project.

**Table 10.** Whole life cycle carbon emission level of the refurbished building and new building.

| Phases | Case Range (tCO$_2$e/m$^2$) | Refurbished Building (tCO$_2$e/m$^2$) | New Building (tCO$_2$e/m$^2$) |
|---|---|---|---|
| Materialized phase | 0.3–0.8 | 0.438 | 0.736 |
| Operational phase | 2–4.5 | 1.94 | 1.94 |
| Whole life cycle | 2–8 | 2.39 | 2.69 |

## 4. Discussion

The above calculation model and the calculation process aim to explore more accurate and convenient solutions. According to the above case study and existing carbon emission calculation cases [6–23,40,41], one of the main difficulties is the uncertainty of data collection and boundary definition process without actual recorded materials. In addition, carbon emission factors have a complex relationship with the socio-economic environment, production technology level, energy structure, ecological environment, and other factors, and are time-sensitive and geographically applicable. Therefore, there may be some differences between the assessment result and the actual situation.

To solve the above two problems, large-scale data input, storage, and processing are needed. A possible solution is the improvement of computing tool performance, during which the authoritative database, including the accumulation and integration of building information, carbon emission factors, and related influencing factors, need to be established and updated. The other possible solution is to build a universal assessment method and mathematical model to simplify the calculation process, which also requires a large amount of data accumulation covering different types of buildings under complex application scenarios in the stage of exploration.

For further study, after the building is put into operation, actual recorded data may be obtained and used to compare with the above result. Based on the comparison, the calculation solutions can be analyzed and optimized.

## 5. Conclusions

The LCA method can be used for the building whole life cycle carbon emission assessment, and the carbon emission factor method is widely applied in the phases of building life cycle.

The main challenge of the calculation process is the boundary selection and inventory collection, especially when carbon emission assessment is needed in the early stage of design and construction, or when the calculation relates to the process of disposal and refurbishment. It is significant to make full and rational use of design documents, standard documents, and related software.

This paper calculated the whole life cycle carbon emission of the Zero-Carbon Pavilion in Shanghai Yangpu Riverside based on the design documents and a simulation model, and then analyzed the potential of energy-saving measures. The results show that the whole life cycle carbon emission of the refurbished building is 2.39 $tCO_2e/m^2$, while the whole life cycle carbon emission of the assumed new building is 2.69 $tCO_2e/m^2$, which are both in a low carbon emission level among the existing cases. The refurbished building saves nearly one-third of the carbon dioxide emissions during the construction phase compared to new construction. The application of a solar photovoltaic system saves one-third of the energy consumption and carbon emission of the building operation.

**Author Contributions:** Conceptualization, Y.P.; methodology, B.L.; software, B.L.; validation, B.L.; formal analysis, B.L.; investigation, B.L.; resources, M.K.; data curation, B.L.; writing—original draft preparation, B.L.; writing—review and editing, Y.P.; visualization, B.L.; supervision, Y.P.; project administration, L.L.; funding acquisition, L.L. All authors have read and agreed to the published version of the manuscript.

**Funding:** This research was funded by the Ministry of Science and Technology (Grant No. G2021133019L) and the National Science Foundation of China (Grant No. 51978481).

**Conflicts of Interest:** The authors declare no conflict of interest.

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
