# Peer review of "Life Cycle Carbon Emission Assessment of Building Refurbishment: A Case Study of Zero-Carbon Pavilion in Shanghai Yangpu Riverside"

_applsci, doi:10.3390/app12199989_

Round 1

Reviewer 1 Report

The authors have applied LCA methodology for a case study. The topic is relevant due to the worldwide requirement to reduce carbon emission.  

The article is well written and shows a clear methodology, however there are a few lingering concerns that requires to be addressed.

Line 61; Three problems are considered  but are not mentioned on the Abstract. It would be nice to show at the Conclusions how does this problems are improved by this work. 

Line 80: I am not sure if "At present" is correct

Line 110: There are some typos including the word "derectely" and  I think "softwares" should be in singular

It would be better if you use the same color coding for the graphs presented later on the paper. 

Line 135: The authors could include more information regarding the Riverside Park. A map and a description of the type of activities that the building hold could provide more understanding for people outside of China. 

Line 145: Please include the translation in English of the labels of the features

Line 261: "Ridiant" typo

Line 304: Capitalize A from Analysis

Line 319: Could you elaborate on the explanation of these other buildings? Could you include additional data to understand where does this data come and what does this represent?? Are these buildings similar to the case study??

Line 326: Could you elaborate on the insights of the study?

How do you explain that the new buidling produces more CO2???! Is it because of the intended use??

Beyond the data I think the authors should add more information that could help future research in this area. What are the strong points of the work? What could be implemented in the future?

Could you tell the readers more regarding are next steps of the research?

Author Response

Dear reviewer,

Thank you so much for your valuable suggestions for our manuscript entitled “‘Life Cycle Carbon Emission Assessment of Building Refurbish ment: A Case Study of Zero-Carbon Pavilion in Shanghai Yangpu Riverside ” . We have made corrections and uploaded the file of the revised manuscript.

Our responses to the comments are listed below.

Point 1: Line 61; Three problems are considered  but are not mentioned on the Abstract. It would be nice to show at the Conclusions how does this problems are improved by this work.

Response 1: Thanks for the valuable suggestion. I will summary these problem into abstract and conclusion.

Point 2: Line 80: I am not sure if "At present" is correct

Response 2: I think the spelling and grammar are correct. If it is necessary, we could change the word to avoid misunderstanding.

Point 3: Line 110: There are some typos including the word "derectely" and  I think "softwares" should be in singular

Response 3: I ‘m sorry for those typos, I will check the text again and correct them.

Point 4: It would be better if you use the same color coding for the graphs presented later on the paper. 

Response 4: Thanks for reminding. I have changed to distinguish the graph items with different patterns. But for the proportional bar chart, there are so much items that different patterns may can not distinguish them clearly, I still use different colors but add borders for the bars.

Point 5: Line 135: The authors could include more information regarding the Riverside Park. A map and a description of the type of activities that the building hold could provide more understanding for people outside of China. 

Response 5: We will add related information including the map and detailed description.

Point 6: Line 145: Please include the translation in English of the labels of the features

Response 6: OK, we will add English translation in the figure.

Point 7: Line 261: "Ridiant" typo

Response 7: I ‘m sorry again for the typos and it has been corrected.

Point 8: Line 304: Capitalize A from Analysis

Response 8: I will correct it.

Point 9: Line 319: Could you elaborate on the explanation of these other buildings? Could you include additional data to understand where does this data come and what does this represent?? Are these buildings similar to the case study??

Response 9: The date is from existing cases with detailed description of calculation process in related papers in China and related papers heve been listed in references[6-23]. Most of these cases have actual recorded materials, which can be used to estimate the level and reliability of the result. What’s more, the carbon emission factors in these cases are from similar references(such as the Building Carbon Emission Calculation Standard issued by Ministry of Housing and Urban-Rural Construction of the People's Republic of China). We will add a brife discription of these cases in the text.

Point 10: Line 326: Could you elaborate on the insights of the study?

Response 10: We will try to add information about the weak points for existing research and put forward to critical comments in the part of discussion.

Point 11: How do you explain that the new buidling produces more CO2???! Is it because of the intended use??

Response 11: It’s because the use of more construction materials compared to the refurbished building. The process of refurbishment is based on the old building constructure, so the carbon emission from the production of materials are reduced.

Point 12: Beyond the data I think the authors should add more information that could help future research in this area. What are the strong points of the work? What could be implemented in the future?

Response 12: We will also try to add the information in the part of discussion.

Point 13: Could you tell the readers more regarding are next steps of the research?

Response 13: OK, We will add the information in discussion too.

Thank you very much.

Sincerely,

Boyang Li

Reviewer 2 Report

The manuscript entitled ‘Life Cycle Carbon Emission Assessment of Building Refurbish ment: A Case Study of Zero-Carbon Pavilion in Shanghai Yangpu Riverside’ is in line with the Applied Sciences journal. It is based on case study analysis. The topic undertaken by the author is important from a practical point of view and up-to-date considering climate changes. The article is quite well planned, but the article requires some minor changes in the content before publication, including:

1) Line 1: select the type of the t

2) Authors: the date for corresponding author are required.

3) Introduction: in lines 31-32, correct the style.

4) Introduction: in lines 41-42, define the period for the unit (a as a year?).

5) Introduction: on line 76, the reference should be directly behind the author name.

6) Introduction: the literature gap should be clarify and the novelty aspects presented in the article should be described.

7) Figures 2.1: please remove double numbers – figures in lines 110 and 141; please use the numeration for tables and content coherent with template for the journal.

8) Line 133 and all article: usage small / capital letters should be corrected, including tables.

9) Table 3.1. (last line) please define the unit in the first column.

10) Figure 3.3. – more detailed comment required, including; value for pumps, efficiency of PV and achieved balance.

11) Figure 3.4. – add information about used data in the text.

12) Figure 3.5. what kind of error is for this number. There is lack of sense give this numbers with 4 places after point.

13) Discussion: add information about the weak points for presented calculations; critical comment

14) References: This part required to be formatted according to editorial template.

Author Response

Dear reviewer,

Thank you so much for your valuable suggestions for our manuscript entitled “‘Life Cycle Carbon Emission Assessment of Building Refurbish ment: A Case Study of Zero-Carbon Pavilion in Shanghai Yangpu Riverside ” .  We have made corrections and uploaded the file of the revised manuscript.

Our responses to the comments are listed below.

Point 1: Line 1: select the type of the t

Response 1: Thank for reminding. I have select the type.

Point 2: Authors: the date for corresponding author are required.

Response 2: Thank for reminding. It has been added.

Point 3: Introduction: in lines 31-32, correct the style.

Response 3: The style have been checked and corrected.

Point 4: Introduction: in lines 41-42, define the period for the unit (a as a year?).

Response 4: Yes, the letter “a” in kWh/((m2·a) means a year.

Point 5: Introduction: on line 76, the reference should be directly behind the author name.

Response 5: Thank for reminding. It have been corrected..

Point 6: Introduction: the literature gap should be clarify and the novelty aspects presented in the article should be described.

Response 6: the literature gap is clarified partly in the above paragraph. We will emphasize  it more clearly in every paragraph.

Point 7: Figures 2.1: please remove double numbers – figures in lines 110 and 141; please use the numeration for tables and content coherent with template for the journal.

Response 7: OK, all the numerations have been corrected.

Point 8: Line 133 and all article: usage small / capital letters should be corrected, including tables.

Response 8: Thank for reminding. All the small / capital letters have been corrected.

Point 9: Table 3.1. (last line) please define the unit in the first column.

Response 9: I'm sorry that the unit was hidden due to the problem of chart format and the format has been corrected.

Point 10: Figure 3.3. – more detailed comment required, including; value for pumps, efficiency of PV and achieved balance.

Response 10: OK, I will summary detailed information in another chart.

Point 11: Figure 3.4. – add information about used data in the text.

Response 11: Brief description have been added in related paragraph. I think if it is better to pay more attention to the proportion and data features to avoid only repeating the date itself.

Point 12: what kind of error is for this number. There is lack of sense give this numbers with 4 places after point.

Response 12: I’m sorry for using the data export from the calculation tool without processing. I will retain two decimal places then.

Point 13: Discussion: add information about the weak points for presented calculations; critical comment

Response 13: OK, I will add and emphasize related information.

Point 14: References: This part required to be formatted according to editorial template.

Response 14: I'm sorry for the negligence. I will format it according to editorial template.

Thank you very much.

Sincerely,

Boyang Li

Reviewer 3 Report

Need to make clearer that paper is a comparison between actual project versus proposed refurbished. Perhaps update title to add “proposed” Paragraph  add more detail about the proposed refurbishment versus new build. Figures 3.4 through 3.6 should also include text clearly describing how the proposed refurbishment is not the actual project.

Author Response

Dear reviewer,

Thanks for your valuable suggestions for our manuscript entitled “Life Cycle Carbon Emission Assessment of Building Refurbish ment: A Case Study of Zero-Carbon Pavilion in Shanghai Yangpu Riverside ”.

I will make clearer discription in the case introduction and emphasis the word "proposed"  in the result analysis .

Thank you so much and the comments are helpful.

Sincerely,

Boyang Li

Round 2

Reviewer 1 Report

Please review the syntaxis and grammar of the new sections of the document. 

Author Response

Dear reviewer,

Thanks for reminding. I have checked and corrected the syntax errors.

Kind regards,

Boyang Li